# WHEN OPTIMIZING $f$-DIVERGENCE IS ROBUST WITH LABEL NOISE

**Jiaheng Wei and Yang Liu**[*]
Department of Computer Science and Engineering
University of California, Santa Cruz
Santa Cruz, CA 95060, USA
{jiahengwei, yangliu}@ucsc.edu

## ABSTRACT

We show when maximizing a properly defined $f$-divergence measure with respect to a classifier's predictions and the supervised labels is robust with label noise. Leveraging its variational form, we derive a nice decoupling property for a family of $f$-divergence measures when label noise presents, where the divergence is shown to be a linear combination of the variational difference defined on the clean distribution and a bias term introduced due to the noise. The above derivation helps us analyze the robustness of different $f$-divergence functions. With established robustness, this family of $f$-divergence functions arises as useful metrics for the problem of learning with noisy labels, which do not require the specification of the labels' noise rate. When they are possibly not robust, we propose fixes to make them so. In addition to the analytical results, we present thorough experimental evidence. Our code is available at `https://github.com/UCSC-REAL/Robust-f-divergence-measures`.

## 1 INTRODUCTION

A machine learning system continuously observes noisy training annotations and it remains a challenge to perform robust training in such scenarios. Earlier and classical approaches rely on estimation processes to understand the noise rate of the labels and then leverage this knowledge to perform label correction (Patrini et al., 2017; Lukasik et al., 2020), or loss correction (Natarajan et al., 2013; Liu & Tao, 2015; Patrini et al., 2017), or both, among many other more carefully designed approaches (please refer to our related work section for more detailed coverage). Recent works have started to propose robust loss functions or metrics that do not require the above estimation (Charoenphakdee et al., 2019; Xu et al., 2019; Liu & Guo, 2020; Cheng et al., 2021). Clear advantages of the latter approaches include their easiness in implementation, as well as their robustness to noisy estimates of the parameters. This work mainly contributes to the second line of studies and aimed to propose relevant loss functions and measures that are inherently robust with label noise.

We start with formulating the problem of maximizing an $f$-divergence defined between a classifier's prediction and the labels:

$$h_f^* = \operatorname*{argmax}_{h} D_f\left(P_{h \times Y} || Q_{h \times Y}\right), \tag{1}$$

where in above $D_f$ is an $f$-divergence function, $P$ and $Q$ are the joint and product (marginal) distribution of the classifier $h$'s predictions on a feature space $X$ and label $Y$. Though optimizing the $f$-divergence measure is in general not the same as finding the Bayes optimal classifiers, we show these measures encourage a classifier that maximizes an extended definition of $f$-mutual information between the classifier's prediction and the true label distribution. We will also provide analysis for when the maximizer of this $f$-divergence coincides with the Bayes optimal classifier.

Building on a careful treatment of its variational form, we then reveal a nice property that helps establish the robustness of the $f$-divergence specified in Eqn. (1): the variational difference term defined with noisy labels is an affine transformation of the clean variational difference, subject to

---

[*]Corresponding author. `yangliu@ucsc.edu`.

an addition of a bias term. Using this result, we analyze under which conditions maximizing an $f$-divergence measure would be robust to label noise. In particular, we demonstrate strong robustness results for Total Variation divergence, identify conditions under which several other divergences, including Jenson-Shannon divergence and Pearson $\mathcal{X}^2$ divergence, are robust. The resultant $f$-divergence functions offer ways to learn with noisy labels, without estimating the noise parameters. As mentioned above, this distinguishes our solutions from a major line of previous studies that would require such estimates. When the $f$-divergence functions are possibly not robust with label noise, our analysis also offers a new way to perform "loss correction". We'd like to emphasize that instead of offering one method/loss/measure, our results effectively offer a family of functions that can be used to perform this noisy training task. Our contributions summarize as follows:

- We show a certain set of $f$-divergence measures that are robust with label noise (some under certain conditions). The corresponding $f$-divergence functions provide the community with robust learning measures that do not require the knowledge of the noise rates.

- When the $f$-divergence measures are possibly not robust with label noise, our analysis provides ways to correct the $f$-divergence functions to offer robustness. This process would require the estimation of the noise rates and our results contribute new ways to leverage existing estimation techniques to make the training more robust.

- We empirically verified the effectiveness of optimizing $f$-divergences when noisy labels present. We opensource our solutions at `https://github.com/UCSC-REAL/Robust-f-divergence-measures`.

## 1.1 RELATED WORKS

The now most popular approach of dealing with label noise is to first estimate the noise transition matrix and then use this knowledge to perform loss or sample correction (Scott et al., 2013; Natarajan et al., 2013; Patrini et al., 2017; Lu et al., 2018; Han et al., 2018; Tanaka et al., 2018; Yao et al., 2020; Zhu et al., 2021). In particular, the surrogate loss (Scott et al., 2013; Natarajan et al., 2013; Scott, 2015; Van Rooyen et al., 2015; Menon et al., 2015) uses the transition matrix to define unbiased estimates of the true losses. Other works include (Sukhbaatar & Fergus, 2014; Xiao et al., 2015), which consider building a neural network to facilitate the learning of noise rates or noise transition matrix. Symmetric loss has been studied and conditions have been identified for when there is no need to estimate noise rate (Manwani & Sastry, 2013; Ghosh et al., 2015; 2017; Van Rooyen et al., 2015; Charoenphakdee et al., 2019). Nonetheless, it remains a challenge to develop training approaches without requiring knowing the noise rates for more generic settings.

More recently, (Zhang & Sabuncu, 2018; Amid et al., 2019) proposed robust losses for neural networks. When noise rates are asymmetric (label class-dependent), (Xu et al., 2019) proposed an information-theoretic loss that is also robust to asymmetric noise rates. There are also some trials on modifying the regularization term to improve generalization ability with the existence of label noise (Jenni & Favaro, 2018; Yi & Wu, 2019), and on providing complementary negative labels (Kim et al., 2019). Peer loss (Liu & Guo, 2020) is a recently proposed loss function that does not require knowing noise rates.

$f$-divergence is a popular information theoretical measure, and has been widely used and studied. Most relevant to us, $f$-GAN was proposed in (Nowozin et al., 2016) to study $f$-divergence in training generative neural samplers. To our best knowledge, ours is the first to study the robustness of $f$-divergence measures in the context of improving the robustness of training with noisy labels.

## 2 LEARNING WITH NOISY LABELS USING $f$-DIVERGENCE

Our solution ties to the definition of $f$-divergence. The $f$-divergence between two distributions $P$ and $Q$ with probability density function $p$ and $q$ being measures for $Z \in \mathcal{Z}^1$ is defined as:

$$D_f(P\|Q) = \int_{\mathcal{Z}} q(Z) f\left(\frac{p(Z)}{q(Z)}\right) dZ .\tag{2}$$

---

[1]We use $Z$ instead of $X$ as conventionally done for a good reason - we will be reserving $X$ to explicitly denote the features.

$f(\cdot)$ is a convex function such that $f(1) = 0$. Examples include KL-divergence when $f(v) = v \log v$ and Total Variation (TV) divergence with $f(v) = \frac{1}{2}|v - 1|$. Other examples can be found in Table 1. Following from Fenchel's convex duality, $f$-divergence admits the following variational form:

$$D_f(P\|Q) = \sup_{g:\mathcal{Z}\to\mathrm{dom}(f^*)} \mathbb{E}_{Z\sim P}\left[g(Z)\right] - \mathbb{E}_{Z\sim Q}\left[f^*(g(Z))\right],$$

where $f^*$ is the Fenchel duality of the function $f(\cdot)$, which is defined as $f^*(u) = \sup_{v\in\mathbb{R}}\{uv - f(v)\}$. We use $\mathrm{dom}(f^*)$ to denote the domain of $f^*$.

We consider the classification problem of learning a classifier $h : \mathcal{X} \to \mathcal{Y}$ that maps features $X \in \mathcal{X}$ to labels $Y \in \mathcal{Y} := \{1, 2, ..., K\}$, where in above $X \times Y$ denote the random variables for features and labels. $X \times Y$ jointly draw from a distribution $\mathcal{D}$. For a clear presentation, we will often focus on presenting the binary classification setting $\mathcal{Y} = \{-1, +1\}$, but most of our core results extend to multi-class classification problems, and we shall provide corresponding justifications.

Instead of having access to sampled training data from $X \times Y$, we consider a setting with noisy labels where the noisy label $\tilde{Y}$ generates according to a transition matrix $T$ defined between $\tilde{Y}$ and the true label $Y$. The $(i, j)$ element of $T$ is defined as $T_{i,j} = \mathbb{P}(\tilde{Y} = j|Y = i)$ where $i, j \in \{1, ..., K\}$. For the ease of presentation, when we present for the binary case, we adopt the following notation: $e_+ := \mathbb{P}(\tilde{Y} = -1|Y = +1)$, $e_- := \mathbb{P}(\tilde{Y} = +1|Y = -1)$, $e_+ + e_- < 1$. Suppose we have access to a noisy training dataset $\{x_n, \tilde{y}_n\}_{n=1}^N$, where $\tilde{y}_n$ generates according to $\tilde{Y}$.

## 2.1 Learning using $D_f$

We will start with presenting our idea of training a classifier using $D_f$ with the clean training data. Then we will proceed to the case with noisy labels. For an arbitrary classifier $h$, let's denote by $P_{h\times Y}$ the joint distribution of $h(X)$ and $Y$:

$$\text{Joint distribution:} \quad P_{h\times Y} := \mathbb{P}(h(X) = y, Y = y'), \ y, y' \in \mathcal{Y}.$$

And we use $Q_{h\times Y}$ to denote the product (marginal) distribution of $h(X)$ and $Y$:

$$\text{Product distribution:} \quad Q_{h\times Y} := \mathbb{P}(h(X) = y) \cdot \mathbb{P}(Y = y'), \ y, y' \in \mathcal{Y}.$$

When it is clear from context we will also shorthand the above two distributions as $P$ and $Q$. We formulate the problem of learning using $f$-divergence as follows: the goal of the learner is to find a classifier $h$ that maximizes the following divergence measure between $P$ and $Q$:

$$\text{Learning using } D_f: \ h_f^* = \underset{h}{\mathrm{argmax}}\, D_f\left(P_{h\times Y}\|Q_{h\times Y}\right) \tag{3}$$

Effectively the goal is to find a classifier that maximizes the divergence between the joint distribution and the product distribution. Define a $f$-mutual information based on $f$-divergence: $M_f(h(X); Y) = D_f\left(P_{h\times Y}\|Q_{h\times Y}\right)$, equivalently the maximization in Eqn. (3) tries to find the classifier that maximizes the $f$-mutual information between a classifier's output distribution and the true label distribution. A notable example is when $f(v) = v \log v$, the corresponding $D_f$ and $M_f$ become the famous KL divergence and the mutual information. It is important to note in general maximizing ($f$-) mutual information between the classifier's predictions and labels does not promise the Bayes optimal classifier $h^* = \mathrm{argmax}_h \mathbb{P}(h(X) = Y)$. Nonetheless, maximizing it often returns a quality one. We provide further analysis in Section 2.2.

**Variational representation** As we mentioned earlier, $f$-divergence admits a variational form which further allows us to focus on maximizing the following variational difference:

$$h_f^* = \underset{h}{\mathrm{argmax}}\, \sup_g \mathbb{E}_{Z\sim P_{h\times Y}}\left[g(Z)\right] - \mathbb{E}_{Z\sim Q_{h\times Y}}\left[f^*(g(Z))\right],$$

where we use $Z$ to shorthand the tuple $[h(X), Y]$. Denote the variational difference as follows:

$$\mathsf{VD}_f(h, g) := \mathbb{E}_{Z\sim P_{h\times Y}}[g(Z)] - \mathbb{E}_{Z\sim Q_{h\times Y}}[f^*(g(Z))]. \tag{4}$$

Let $g^*$ be the corresponding optimal variational function $g$ for $\mathsf{VD}_f(h, g)$. This variational form allows us to use a training dataset $\{(x_n, y_n)\}_{n=1}^N$ to perform the above maximization problem listed in Eqn. (3) (Nowozin et al., 2016). A list of $f$-divergence functions together with the optimal variational/conjugate functions $g/f^*$ is summarized in Table 1.

| Name | $D_f(P||Q)$ | $g^*$ | $\text{dom}_{f^*}$ | $f^*(u)$ |
|---|---|---|---|---|
| Total Variation | $\int \frac{1}{2}|p(z) - q(z)|dz$ | $\frac{1}{2}\,\text{sign}\,\frac{p(z)}{q(z)} - 1$ | $u \in [-\frac{1}{2}, \frac{1}{2}]$ | $u$ |
| Jenson-Shannon | $\frac{1}{2}\int p(z)\log\frac{2p(z)}{p(z)+q(z)} + q(z)\log\frac{2q(z)}{p(z)+q(z)}dz$ | $\log\frac{2p(z)}{p(z)+q(z)}$ | $u < \log 2$ | $-\log(2 - e^u)$ |
| Pearson $\mathcal{X}^2$ | $\int \frac{(q(z) - p(z))^2}{p(z)}dz$ | $2\left(\frac{p(z)}{q(z)} - 1\right)$ | $\mathbb{R}$ | $\frac{1}{4}u^2 + u$ |
| KL | $\int p(z)\log\frac{p(z)}{q(z)}dx$ | $1 + \log\frac{p(z)}{q(z)}$ | $\mathbb{R}$ | $e^{u-1}$ |

Table 1: $D_f$s, optimal variational $g$ ($g^*$), conjugate functions ($f^*$). A more complete table, including Jeffrey, Squared Hellinger, Neyman $\mathcal{X}^2$, Reverse KL, is provided in the Appendix.

## 2.2 HOW GOOD IS $h_f^*$?

As we mentioned earlier, maximizing our defined $f$-divergence measures (or maximizing the $f$-mutual information) between the classifier's predictions and labels is not always returning the Bayes optimal classifier. However, for a binary classification problem, we prove below that with balanced dataset, maximizing Total Variation (TV) divergence returns the Bayes optimal classifier:

**Theorem 1.** *For TV, when $\mathbb{P}(Y = +1) = \mathbb{P}(Y = -1)$ (balanced), $h_f^*$ is the Bayes optimal classifier.*

**Remark 2.** *The above theorem extends to the multi-class setting when we restrict attentions to confident classifiers. See Appendix for details.*

The above observation is not easily true for other $f$-divergence. Nonetheless, denote by $Y^*(X = x)$ the Bayes optimal label for an instance $x$: $Y^*(X = x) = \text{argmax}_y \mathbb{P}(Y = y|X = x)$. Denote by $P_{h \times Y^*}, Q_{h \times Y^*}$ the joint and product distribution $P, Q$ defined w.r.t. $h(X)$ and $Y^*$. We prove:

**Theorem 3.** *When $\mathbb{P}(Y^* = +1) = \mathbb{P}(Y^* = -1)$ (balanced), maximizing $D_f(P_{h \times Y^*}||Q_{h \times Y^*})$ returns the Bayes optimal classifier, if $f(v)$ is monotonically increasing in $|v - 1|$ on $\text{dom}(f)$.*

For example, Pearson $\mathcal{X}^2$ ($f(v) = (v-1)^2$) satisfies the monotonicity condition. In practice, when the label distribution $\mathbb{P}(Y|X = x)$ has small uncertainties, the ground truth labels are approximately equivalent to the Bayes optimal label. Therefore, the above theorem implies that maximizing $D_f(P_{h \times Y}||Q_{h \times Y})$ is also likely to return a high-quality classifier for other $f$-divergences.

## 2.3 LEARNING WITH NOISY LABELS

Consider an arbitrary classifier $h$. Denote by $\tilde{P}_{h \times \tilde{Y}}$ the joint distribution of $h(X)$ and $\tilde{Y}$:

$$\text{Joint noisy distribution:} \quad \tilde{P}_{h \times \tilde{Y}} := \mathbb{P}(h(X) = y, \tilde{Y} = y'), \; y, y' \in \mathcal{Y}.$$

Similarly, we use $\tilde{Q}_{h \times \tilde{Y}}$ to denote the product (marginal) distribution of $h(X)$ and $\tilde{Y}$:

$$\text{Product noisy distribution:} \quad \tilde{Q}_{h \times \tilde{Y}} := \mathbb{P}(h(X) = y) \cdot \mathbb{P}(\tilde{Y} = y'), \; y, y' \in \mathcal{Y}.$$

When it is clear from context, we shorthand using $\tilde{P}, \tilde{Q}$. We are interested in understanding the robustness in maximizing $D_f(\tilde{P}_{h \times \tilde{Y}}||\tilde{Q}_{h \times \tilde{Y}})$. Using training samples $\{x_n, \tilde{y}_n\}_{n=1}^N$, there exists algorithms to compute the gradient of $D_f$ leveraging its variational form (Nowozin et al., 2016), such that one can apply gradient descent or ascent to optimize it. We provide details in Section 5.

## 3 VARIATIONAL DIFFERENCE WITH NOISY LABELS

For an arbitrary $g$, we define the variational difference term w.r.t. the noisy label as follows:

$$\widetilde{\text{VD}}_f(h, g) := \mathbb{E}_{\tilde{Z} \sim \tilde{P}_{h \times \tilde{Y}}}\left[g(\tilde{Z})\right] - \mathbb{E}_{\tilde{Z} \sim \tilde{Q}_{h \times \tilde{Y}}}\left[f^*(g(\tilde{Z}))\right] \tag{5}$$

where we use $\tilde{Z}$ to denote $[h(X), \tilde{Y}]$. Denote by $\tilde{g}^*$ the corresponding optimal variational function $g$ for $\widetilde{\text{VD}}_f(h, g)$. In this section, we show that the variational difference term under noisy labels is closely related to the variational difference term defined on the clean distributions $P, Q$. Define the following quantity: $\Delta_f^y(h, g) := \mathbb{E}_X[g(h(X), y)] - \mathbb{E}_X[f^*(g(h(X), y))]$, for example

$\Delta_f^{+1}(h,g) := \mathbb{E}_X[g(h(X), +1)] - \mathbb{E}_X[f^*(g(h(X), +1))]$. For a binary classification problem, further denote by $\mathsf{Bias}_f(h,g) := e_+ \cdot \Delta_f^{-1}(h,g) + e_- \cdot \Delta_f^{+1}(h,g)$. We derive the following fact:

**Theorem 4.** *For binary classification, the variational difference between the noisy distributions $\tilde{P}$ and $\tilde{Q}$ relates to the one defined on the clean distributions in the following way:*

$$\widetilde{\mathsf{VD}}_f(h,g) = (1 - e_+ - e_-)\,\mathsf{VD}_f(h,g) + \mathsf{Bias}_f(h,g) \tag{6}$$

The above decoupling result is inspiring: $\mathsf{Bias}_f(h,g)$ can be viewed as the additional bias term introduced by label noise. If this term has negligible effect in the maximization problem, maximizing the noisy variational difference term will be equivalent to maximizing $(1 - e_+ - e_-) \cdot \mathsf{VD}_f(h,g)$, and therefore the clean variational difference term. If the above is true, we have established the robustness of the corresponding $f$-divergence. This result also points out that when the effects from the bias term are non-negligible, finding ways to counter the additional bias term will help us retain the robustness of $D_f$ measures. Next we show that Theorem 4 extends to the multi-class setting under two broad families of noise rate models, both covering the binary setting as a special case.

**Multi-class extension of Theorem 4: uniform off-diagonal case** We first consider the following transition matrix: uniform off-diagonal transition matrix, where $e_j = T_{i,j}, \forall i \neq j$, that is any other classes $i \neq j$ has the same chance of being flipped to class $j$. The diagonal entry $T_{i,i}$ (chance of a correct label) becomes $1 - \sum_{j \neq i} e_j$. We further require that $\sum_j e_j < 1$. Note that the binary noise rate model is easily a uniform off-diagonal transition matrix.

**Theorem 5.** *[Multi-class] For uniform off-diagonal noise transition model, the noisy variational difference term relates to the clean one in the following way:*

$$\widetilde{\mathsf{VD}}_f(h,g) = \left(1 - \sum_{j=1}^K e_j\right) \cdot \mathsf{VD}_f(h,g) + \sum_{j=1}^K e_j \cdot \Delta_f^j(h,g) \tag{7}$$

If we define $\mathsf{Bias}_f(h,g) := \sum_{j=1}^K e_j \cdot \Delta_f^j(h,g)$, we reproduced the results in Theorem 4: for binary case, relabel class $1 \to +1, 2 \to -1$. Then $e_1 := \mathbb{P}(\tilde{Y} = +1|Y = -1) = e_-, e_2 := \mathbb{P}(\tilde{Y} = -1|Y = +1) = e_+$. Another case of noise model we consider is *sparse noise*. Mathematically, assume $K$ is an even number, sparse noise model specifies $\frac{K}{2}$ disjoint pairs of classes $(i_c, j_c)$ where $c \in [\frac{K}{2}]$ and $i_c < j_c$. The labels flip between each pair. We provide details in the Appendix.

## 4 WHEN $D_f$ IS ROBUST WITH LABEL NOISE

Denote by $\mathcal{H}$ an arbitrary hypothesis space for training a candidate classifier $h$. We will focus on $\mathcal{H}$ throughout this section, and with abusing notation a bit, let $h_f^* = \mathrm{argmax}_{h \in \mathcal{H}} D_f(P_{h \times Y} \| Q_{h \times Y})$. We first define formally what we mean by robustness of $D_f(P_{h \times Y} \| Q_{h \times Y})$.

**Definition 1.** $D_f(P_{h \times Y} \| Q_{h \times Y})$ *is $\mathcal{H}$-robust if* $h_f^* = \mathrm{argmax}_{h \in \mathcal{H}} D_f(\tilde{P}_{h \times \tilde{Y}} \| \tilde{Q}_{h \times \tilde{Y}})$.

The above definition is stating that the label noise does not disrupt the optimality of $h_f^*$ when maximizing $D_f(\tilde{P}_{h \times \tilde{Y}} \| \tilde{Q}_{h \times \tilde{Y}})$ instead of $D_f(P_{h \times Y} \| Q_{h \times Y})$.

### 4.1 IMPACT OF THE BIAS TERMS

In this section, we take a closer look at the $\mathsf{Bias}$ terms and argue that they have diminishing effects as compared to the $\mathsf{VD}$ terms when label noise increases. Recall $g^*, \tilde{g}^*$ are the corresponding optimal variational functions for $\mathsf{VD}_f(h,g)$ and $\widetilde{\mathsf{VD}}_f(h,g)$.

**Total variation (TV)** For TV, since $f(v) = \frac{1}{2}|v - 1|, f^*(u) = u$, we immediately have $\forall y'\ g(h = y', y) - f^*(g(h = y', y)) = 0$ and therefore $\Delta_f^y(h,g) = \mathbb{E}_X[g(h(X), y)] - \mathbb{E}_X[f^*(g(h(X), y))] \equiv 0, \forall y$, and further $\mathsf{Bias}_f(h,g) \equiv 0$. This fact helps establish the robustness of TV divergence measure (Theorem 7).

**Other divergences** The above nice property generally does not hold for other $f$-divergence functions. Next we focus on the binary classification setting and prove the following lemma:

**Lemma 1.** *For $f$-divergence listed in Table 6 (Appendix),* $\textsf{Bias}_f(h, \tilde{g}^*) = O\left((1 - e_+ - e_-)^2\right)$.

Note the variational form will be used when optimizing $D_f(\tilde{P}_{h \times \tilde{Y}} || \tilde{Q}_{h \times \tilde{Y}})$ (and therefore we will be using $\tilde{g}^*$). This lemma simplifies Eqn. 4 to $\widetilde{\textsf{VD}}_f(h, g) \propto \textsf{VD}_f(h, g) + O(1 - e_+ - e_-)$. Since $0 < 1 - e_+ - e_- \leq 1$, when the noise rate $e_+ + e_-$ is high, the effect of $\textsf{Bias}$ term diminishes. When the $\textsf{Bias}$ term becomes negligible, we will have $\widetilde{\textsf{VD}}_f(h, g) \propto \textsf{VD}_f(h, g)$ if $e_+ + e_- \to 1$, establishing the fact that optimizing $\widetilde{\textsf{VD}}_f(h, g)$ is approximately the same as optimizing $\textsf{VD}_f(h, g)$.

## 4.2 How robust are $D_f$s?

We first prove the following result:

**Theorem 6.** *$D_f$ is $\mathcal{H}$-robust when $\textsf{Bias}_f(h, g)$ satisfies either of the following conditions: (I) $\forall h \in \mathcal{H}$, $\textsf{Bias}_f(h, g) \equiv const.$; (II) $\forall h \in \mathcal{H}, h \neq h_f^*$, $\textsf{Bias}_f(h, \tilde{g}^*) \leq \textsf{Bias}_f(h_f^*, g^*)$.*

Theorem 6 gives sufficient conditions when the $\textsf{Bias}$ term does not get in the way of reaching the optimality $h_f^*$. Intuitively, when $\textsf{Bias}_f(h_f^*, g^*)$ is an upper bound of $\textsf{Bias}_f(h, \tilde{g}^*)$, the $\textsf{Bias}$ term will not interfere with the convergence of the $\textsf{VD}$ term. Next we provide specific examples of $f$-divergence functions that would satisfy these conditions.

**Total Variation (TV) is robust** For TV, the fact that $\Delta_f^y(h, g) \equiv 0$ allows us to prove:

**Theorem 7.** *For TV divergence, $\textsf{Bias}_f(h, g) \equiv const.$ and $D_f(P_{h \times Y} || Q_{h \times Y})$ is $\mathcal{H}$-robust with label noise for any arbitrary hypothesis space $\mathcal{H}$.*

This result establishes TV as a strong measure that does not require specifying the noise rates.

**Divergences that are conditionally robust** Other divergences functions do not enjoy the above nice property as TV has. The robustness of these functions need more careful analysis. Define the following measures that capture the degree a classifier fits to a particular label distribution:

**Definition 2.** *The fitness of $h$ to $R \in \{Y, \tilde{Y}\}$ is defined as $\textsf{FIT}(h = y, R = y') := \frac{\mathbb{P}(h(X)=y|R=y')}{\mathbb{P}(h(X)=y)}$.*

$\textsf{FIT}$ measures capture the degree of fit of the classifier to the corresponding label distribution. A high $\textsf{FIT}(h = y, \tilde{Y} = y)$ (same label) indicates a potential overfit to the noisy label. Denote by

$$\mathcal{H}^* := \{h \in \mathcal{H} : \min_y \textsf{FIT}(h = y, \tilde{Y} = y) \geq \max_y \textsf{FIT}(h_f^* = y, Y = y) \geq 1\} \cup \{h_f^*\}$$

The 1 in the "$\geq 1$" above corresponds to the $\textsf{FIT}$ for a random classifier. $\mathcal{H}^*$ contains the classifiers that are likely to overfit to the noisy labels. We argue, and also as observed in training, that $\mathcal{H}^*$ is the set of classifiers the training should avoid converging to, especially when the training only sees noisy labels. Suppose $\mathbb{P}(Y = +1) = \mathbb{P}(Y = -1)$ (balanced clean labels) and $e_+ = e_-$ (symmetric noise rate), we have the following theorem for binary classification:

**Theorem 8.** *$f$-divergences listed in Table 6 (Appendix, except for Jeffrey) are $\mathcal{H}^*$-robust.*

## 4.3 Making $D_f$ measures robust to label noise

For the general case, to further improve robustness of $D_f$ measures, we will need to estimate the noise rates (e.g., $e_+, e_-$) and then subtract $\textsf{Bias}_f(g, h)$ from the noisy variational difference term to correct the bias introduced by the noisy labels. As a corollary of Theorem 4 we have:

**Corollary 1.** *Maximizing the following bias-corrected $\widetilde{\textsf{VD}}_f(h, g)$ defined over $\tilde{P}$ and $\tilde{Q}$ leads to $h_f^*$*

$$h_f^* = \operatorname*{argmax}_{h \in \mathcal{H}} \sup_g \ \mathbb{E}_{\tilde{Z} \sim \tilde{P}_{h \times \tilde{Y}}}\left[g(\tilde{Z})\right] - \mathbb{E}_{\tilde{Z} \sim \tilde{Q}_{h \times \tilde{Y}}}\left[f^*(g(\tilde{Z}))\right] - \textsf{Bias}_f(h, g) \ .$$

By removing the $\textsf{Bias}_f$ term, maximizing $\mathbb{E}_{\tilde{Z} \sim \tilde{P}}[g(\tilde{Z})] - \mathbb{E}_{\tilde{Z} \sim \tilde{Q}}[f^*(g(\tilde{Z}))]$ becomes the same with maximizing the divergence defined on the clean distribution $(1 - \sum_{j=1}^K e_j) \cdot \textsf{VD}_f(h, g)$. The Corollary follows trivially from this fact. The calculation of the $\textsf{Bias}$ terms will require the inputs of

noise rates. Our work does not intend to particularly focus on noise rate estimation. But rather, we can leverage the existing results in performing efficient noise rate estimation. There are existing literature on estimating noise rates (noise transition matrix) which can be implemented without the need of ground truth labels. For interested readers, please refer to (Liu & Tao, 2015; Menon et al., 2015; Harish et al., 2016; Patrini et al., 2017; Arazo et al., 2019; Yao et al., 2020; Zhu et al., 2021). We will test the effectiveness of this bias correction step in Section 5.

## 5 EXPERIMENTS

In this section, we validate our analysis of $D_f$ measures' robustness via a set of empirical evaluations on 5 datasets: MNIST (LeCun et al. (1998)), Fashion-MNIST (Xiao et al. (2017)), CIFAR-10 and CIFAR-100 (Krizhevsky et al. (2009)), and Clothing1M (Xiao et al. (2015)). Omitted experiment details are available in the appendix.

**Baselines** We compare our approach with five baseline methods: **Cross-Entropy (CE)**, **Backward (BLC) and Forward Loss Correction (FLC)** methods as introduced in (Patrini et al., 2017), the **determinant-based mutual information (DMI)** method introduced in (Xu et al., 2019) and **Peer-Loss (PL)** functions in (Liu & Guo, 2020). BLC and FLC methods require estimating the noise transition matrix. DMI and PL are approaches that do not require such estimation.

**Noise model** We test three types of noise transition models: uniform noise, sparse noise, and random noise. All details of the noise are in the Appendix. Here we briefly overview them. The uniform and sparse noise are as specified at the end of Section 3 for which our theoretical analyses mainly focus on. The noise rates of low-level uniform noise and sparse noise are both approximately 0.2 (the average probability of a label being wrong). The high-levels are about 0.55 and 0.4 respectively. In the random noise setting, each class randomly flips to one of 10 classes with probability $p$ (Random $p$). For CIFAR-100, the noise rate of uniform noise is about 0.25. The sparse label noise is generated by randomly dividing 100 classes into 50 pairs, and the noise rate is about 0.4.

**Optimizing** $D_f(\tilde{P}_{h \times \tilde{Y}} || \tilde{Q}_{h \times \tilde{Y}})$ **using noisy samples** With the noisy training dataset $\{x_n, \tilde{y}_n\}_{n=1}^N$, we optimize $D_f(\tilde{P}_{h \times \tilde{Y}} || \tilde{Q}_{h \times \tilde{Y}})$ using gradient ascent of its variational form. Sketch is given in Algorithm 1. For the bias correction version of our algorithm, the gradient will simply include the $\nabla \mathsf{Bias}_f(h, g)$. The variational function $\tilde{g}^*$ can be updated progressively or can be fixed beforehand using an approximate activation function for each $f$ (see e.g., (Nowozin et al., 2016)).

---

**Algorithm 1** Maximizing $D_f$ measures: one step gradient

1: **Inputs**: Training data $\{(x_n, \tilde{y}_n)\}_{n=1}^N$, $f$, variational function $\tilde{g}^*$, conjugate $f^*$, classifier $h_t$.
2: Randomly sample three mini-batches $\{(x_n, \tilde{y}_n)\}_{n=1}^B$, $\{(x_n^\dagger, \tilde{y}_n^\dagger)\}_{n=1}^B$, $\{(x_n^\diamond, \tilde{y}_n^\diamond)\}_{n=1}^B$ from $\{(x_n, \tilde{y}_n)\}_{n=1}^N$. $\{(x_n, \tilde{y}_n)\}_{n=1}^B$: simulate samples $\sim \tilde{P}$; $\{(x_n^\dagger, \tilde{y}_n^\diamond)\}_{n=1}^B$ to simulate $\tilde{Q}$.
3: Use $h_{t,x_n}[\tilde{y}_n]$ to denote model prediction on $x_n$ for label $\tilde{y}_n$, $\widetilde{\mathbb{E}}_{\{(x_n, \tilde{y}_n)\}_{n=1}^B}$, $\widetilde{\mathbb{E}}_{\{(x_n^\dagger, \tilde{y}_n^\diamond)\}_{n=1}^B}$ to denote the empirical sample mean calculated using the mini-batch data.
4: At step $t$, update $h_t$ by ascending its stochastic gradient with learning rate $\eta_t$:

$$h_{t+1} := h_t + \eta_t \cdot \nabla_{h_t} \left[ \widetilde{\mathbb{E}}_{\{(x_n, \tilde{y}_n)\}_{n=1}^B} [\tilde{g}^* (h_{t,x_n}[\tilde{y}_n])] - \widetilde{\mathbb{E}}_{\{(x_n^\dagger, \tilde{y}_n^\diamond)\}_{n=1}^B} [f^* (\tilde{g}^*(h_{t,x_n^\dagger}[\tilde{y}_n^\diamond]))] \right].$$

Tips: In practice, we suggest (also implemented in our experiments) using the fixed form of $\tilde{g}^*$ which appears as $g_f(v)$ in Table 6 (appendix).

---

### 5.1 HOW GOOD IS $h_f^*$ ON CLEAN DATA

As a supplementary of Section 2.2, we validate the quality of $h_f^*$ on clean dataset of MNIST, Fashion MNIST, CIFAR-10 and CIFAR-100. In experiments, since the estimation of product noisy distribution are unstable when trained on CIFAR-100 training dataset, we use CE as a warm-up (120 epochs) and then switch to train with $D_f$ measures. For other datasets, we train with $D_f$ measures without the warm-up stage. Results in Table 2 demonstrate that optimizing $f-$divergence on clean dataset returns a high-quality $h_f^*$ by referring to the performance of CE. Even though $D_f$ measures

can't outperform CE on clean dataset, we do observe that the gap between CE and $D_f$ measures are negligible, for example, the largest gap of Total-Variation (TV) is only $0.81\%$ among four datasets.

| Dataset | CE | TV | Gap | J-S | Gap | KL | Gap |
|---|---|---|---|---|---|---|---|
| MNIST | 99.39(99.38±0.01) | 99.37(99.34±0.02) | **-0.04** | 99.35(99.31±0.04) | **-0.07** | 99.31(99.21±0.06) | **-0.17** |
| Fashion MNIST | 90.44(90.34±0.12) | 89.98(89.94±0.06) | **-0.40** | 90.40(90.17±0.24) | **-0.17** | 90.19(89.96±0.14) | **-0.38** |
| CIFAR-10 | 93.58(93.47±0.08) | 92.80(92.66±0.13) | **-0.81** | 92.35(92.23±0.07) | -1.24 | 90.55(90.38±0.15) | -3.09 |
| CIFAR-100 | 73.47(73.39±0.05) | 73.43(73.39±0.06) | **0.00** | 73.47(73.26±0.17) | **-0.13** | 73.33(73.16±0.10) | **-0.23** |

Table 2: Experiment results comparison on clean datasets: We report the maximum accuracy of CE and each $D_f$ measures along with (mean $\pm$ standard deviation); Gap: mean performance comparison w.r.t. CE. Numbers highlighted in **blue** indicate the gap is less than 1%.

## 5.2 ROBUSTNESS OF $D_f$ MEASURES

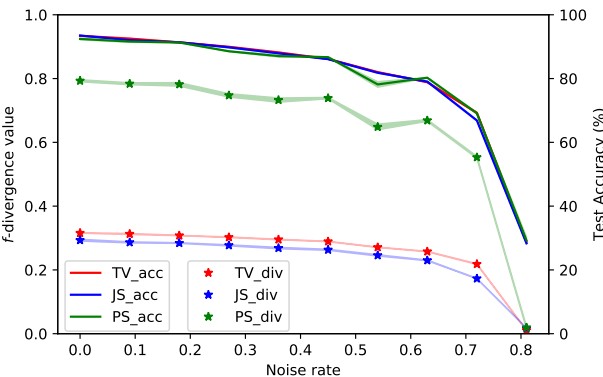

Figure 1: Robustness of TV, JS, PS divergences.

As a demonstration, we apply the uniform noise model to CIFAR-10 dataset to test the robustness of three $D_f$ measures: Total-Variation (TV), Jenson-Shannon (JS) and Pearson (PS). We trained models with $D_f$ measures using Algorithm 1 on 10 noise settings with an increasing noise rate from 0% to approximately 81%. The visualization of the $D_f$ values and accuracy w.r.t. noise rates are shown in Figure 1. Both the $D_f$ values and test accuracy are calculated on the reserved clean test data. We observe that almost all $D_f$ measures are robust to noisy labels, especially when the percentage of noisy labels is not overwhelmingly large, e.g., $\leq 70\%$. Note that the curves for other $f$-divergences are almost the same as the curve of total variation (TV), which is proved to be robust theoretically. This partially validates the analytical evidences we provided for the robustness of other $f$-divergences in Section 4.1 and 4.2.

## 5.3 PERFORMANCE EVALUATION AND COMPARISON

From Table 3, several $D_f$ measures arise as competitive solutions in a variety of noise scenarios. Among the proposed $f$-divergences, Total Variation (TV) has been consistently ranked as one of the top performing method. This aligns also with our analyses that TV is inherently robust. For most settings, the presented $f$-divergences outperformed the baselines we compare to, while they fell short to DMI (once) and Peer Loss (5 times) on several cases, particularly when the noise is sparse and high. The sparse high noise setting tends to be a challenging setting for all methods. We conjecture this is because sparse high noise setting creates a highly imbalanced dataset, model training is more likely to converge to a "sub-optimal" early in the training process. It is also possible that with sparse noise, the impact of Bias terms becomes non-negligible. We do observe better performances with very careful and intensive hyper-parameter tuning, but the results are not confident and we chose to not report it. Fully understanding the limitation of our approach in this setting remains an interesting on-going investigation.

In Table 4 (full details on MNIST and Fashion MNIST can be found in Appendix), we use noise transition estimation method in (Patrini et al. (2017)) to estimate the noise rate. The estimates help

us define the bias term and perform bias correction for $D_f$ measures. We observe that while adding bias correction can further improve the performance of several divergence functions (Gap being positive), the improvement or difference is not significant. This partially justified our analysis of the bias term, especially when the noise is dense and high (uniform and random high).

| Dataset | Noise | CE | BLC | FLC | DMI | PL | TV | J-S | KL |
|---|---|---|---|---|---|---|---|---|---|
| MNIST | Sparse, Low | 97.21 | 95.23 | 97.37 | 97.76 | 98.59 | **99.23(99.11±0.08)** | **99.15(99.03±0.09)** | **99.21(99.15±0.05)** |
| | Sparse, High | 48.55 | 55.86 | 49.67 | 49.61 | **60.27** | 58.27(54.72±4.36) | 58.93(55.80±1.93) | 49.24(49.17±0.06) |
| | Uniform, Low | 97.14 | 94.27 | 95.51 | 97.72 | 99.06 | **99.23(99.17±0.05)** | **99.1(99.08±0.04)** | **99.13(99.06±0.07)** |
| | Uniform, High | 93.25 | 85.92 | 87.75 | 95.50 | 97.77 | **98.09(97.96±0.13)** | 97.86(97.71±0.10) | **98.14(97.88±0.18)** |
| | Random (0.2) | 98.26 | 97.46 | 97.61 | 98.82 | 99.25 | 99.26(99.19±0.05) | **99.29(99.27±0.02)** | 99.26(99.19±0.06) |
| | Random (0.7) | 97.00 | 93.52 | 87.74 | 95.47 | 98.52 | **98.81(98.73±0.06)** | 98.72(98.63±0.08) | **98.76(98.65±0.10)** |
| Fashion MNIST | Sparse, Low | 84.36 | 86.02 | 88.15 | 85.65 | 88.32 | **89.74(89.34±0.33)** | 88.80(88.79±0.01) | **89.77(89.42±0.34)** |
| | Sparse, High | 43.33 | 46.97 | 47.63 | 47.16 | **51.92** | 45.66(45.22±0.26) | 47.46(46.39±0.70) | 38.96(38.90±0.06) |
| | Uniform, Low | 82.98 | 84.48 | 86.58 | 83.69 | **89.31** | 89.00(88.75±0.16) | 88.58(88.46±0.18) | 88.32(88.16±0.11) |
| | Uniform, High | 79.52 | 78.10 | 82.41 | 77.94 | 84.69 | **85.58(85.07±0.31)** | **85.62(85.39± 0.33)** | **85.69(85.43±0.30)** |
| | Random (0.2) | 85.47 | 83.40 | 77.61 | 86.21 | 89.78 | **90.22(90.09±0.19)** | 89.73(89.43±0.24) | 89.24(89.05±0.14) |
| | Random (0.7) | 82.05 | 78.41 | 73.42 | 80.89 | 87.22 | 86.69(86.49±0.16) | **87.79(87.33±0.29)** | 87.06(87.00±0.06) |
| CIFAR-10 | Sparse, Low | 87.20 | 72.96 | 76.17 | **92.32** | 91.35 | 91.81(91.56±0.16) | 91.49(91.43±0.08) | 91.62(91.32±0.31) |
| | Sparse, High | 61.81 | 56.30 | 66.12 | 27.94 | **69.70** | 63.96(62.25±1.00) | 67.33(65.27±1.34) | 46.55(46.43±0.08) |
| | Uniform, Low | 85.68 | 72.73 | 77.12 | 90.39 | 91.70 | **92.10(92.01±0.09)** | 91.52(91.47±0.08) | **92.26(92.08±0.12)** |
| | Uniform, High | 71.38 | 54.41 | 64.22 | 82.68 | 83.42 | **85.56(85.44±0.08)** | 84.49(84.35±0.13) | 84.36(84.19±0.13) |
| | Random (0.5) | 78.40 | 59.31 | 68.97 | 85.06 | 86.47 | 87.28(87.03±0.17) | 86.92(86.80±0.10) | 86.93(86.85±0.11) |
| | Random (0.7) | 68.26 | 38.59 | 54.39 | 77.91 | 57.81 | **80.59(80.45±0.10)** | 80.50(80.27±0.15) | 78.93(78.59±0.30) |
| CIFAR-100 | Uniform | 63.87 | 51.40 | 60.04 | 64.39 | 67.94 | **69.15(68.90±0.17)** | 69.13(68.80±0.21) | 68.79(68.60±0.11) |
| | Sparse | 40.45 | 36.57 | 43.39 | 40.53 | **44.25** | 42.45(38.06±2.82) | 38.09(38.00±0.08) | 37.74(37.63±0.08) |
| | Random (0.2) | 65.84 | 61.21 | 61.52 | 66.23 | 62.92 | **70.43(70.22±0.13)** | 70.40(70.12±0.21) | 70.28(70.06±0.14) |
| | Random (0.5) | 56.92 | 22.21 | 55.88 | 56.06 | 49.62 | **62.14(61.89±0.18)** | 61.58(61.15±0.27) | 61.68(61.49±0.13) |

Table 3: Experiment results comparison (w/o bias correction): The best performance in each setting is highlighted in **blue**. We report the maximum accuracy of each $D_f$ measures along with (mean $\pm$ standard deviation). All $f$-divergences will be highlighted if their mean performances are better (or no worse) than all baselines we compare to. A supplementary table including Pearson $\mathcal{X}^2$ and Jeffrey (JF) is attached in Table 7 (Appendix).

| Noise | J-S | Gap | PS | Gap | KL | Gap | JF | Gap |
|---|---|---|---|---|---|---|---|---|
| Sparse, Low | 91.23(90.93±0.34) | -0.26 | 91.48(91.12±0.42) | **+0.08** | 91.73(91.57±0.18) | **+0.11** | 91.45(91.18±0.21) | -0.10 |
| Sparse, High | 46.45(46.31±0.14) | -20.88 | 46.31(45.90±0.44) | -0.05 | 46.59(46.52±0.05) | **+0.04** | 46.25(45.77±0.50) | **+0.04** |
| Uniform, Low | **92.16(92.09±0.09)** | **+0.64** | **92.25(92.13±0.09)** | -0.12 | 90.92(90.84±0.10) | -1.34 | **92.19(92.10±0.08)** | **+0.02** |
| Uniform, High | 84.31(84.13±0.10) | -0.18 | **83.79(83.61±0.12)** | **+0.18** | 83.98(83.79±0.12) | -0.38 | **83.93(83.62±0.22)** | **+0.13** |

Table 4: $D_f$ measures with bias correction on CIFAR-10: Numbers highlighted in **blue** indicate better than all baseline methods; Gap: relative performance w.r.t. their version w/o bias correction (Table 3); those in **red** indicate better than w/o bias correction.

**Clothing1M**   Clothing1M is a large-scale clothes dataset with comprehensive annotations and can be categorized as a feature-dependent human-level noise dataset. Although this noise setting does not exactly follow our assumption, we are interested in testing the robustness of our $f$-divergence approaches. Experiment results in Table 5 demonstrate the robustness of the $D_f$ measures. TV and KL divergences have outperformed other baseline methods.

| Dataset | Noise | CE | BLC | FLC | DMI | PL | T-V | J-S | Pear | KL | Jeffrey |
|---|---|---|---|---|---|---|---|---|---|---|---|
| Clothing1M | Human Noise | 68.94 | 69.13 | 69.84 | 72.46 | 72.60 | **73.09** | 72.32 | 72.22 | **72.65** | 72.46 |

Table 5: Experiment results comparison on Clothing1M dataset.

# 6 CONCLUSION

In this paper, we explore the robustness of a properly defined $f$-divergence measure when used to train a classifier in the presence of label noise. We identified a set of nice robustness properties for a family of $f$-divergence functions. We also experimentally verified our findings. Our work primarily contributed to the problem of learning with noisy labels without requiring the knowledge of noise rate. Beyond this noisy learning problem, the derivation and analysis might be useful for understanding the robustness of $f$-divergences for other learning tasks.

**Acknowledgement**   This work is partially supported by the National Science Foundation (NSF) under grant IIS-2007951 and the Office of Naval Research under grant N00014-20-1-22.

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
