# OpenReview forum: "When Optimizing  $f$-Divergence is Robust with Label Noise"
_ICLR.cc/2021/Conference — ICLR 2021 Poster_

### Official Review · AnonReviewer3 · 2020-10-27
**f-divergence is robust to label noise**

**Rating:** 7
**Confidence:** 3

**Review:**

The authors study optimising f-divergence for supervised learning in the context of label noise. They show robustness to noise for certain families of f-divergence functions and devise a technique for minimising bias for other f-divergence functions. They evaluate their method against pre-existing methods as baselines on a variety of learning tasks including simulation and real-world data. There are some minor issues to address but overall a good submission.

# pros

- Easy to read paper, the authors guide the reader well through the theory
- proofs are sound
- good set of experiments
- results are interesting and applicable in many domains, of broad interest to the community

# cons

- The experiments are not described in sufficient detail. It is unclear what h is used for experiments and what variational function was used exactly. The authors state g* can be updated progressively or fixed beforehand, but neglect to say what they did for experiments.
- The authors mention the BLC and FLC methods require estimating the noise transition matrix: how was this done?

# minor things

- typo on page 3: f(v)= log v for KL divergence

---

> ### Author Response · Authors · 2020-11-13
> **Response to Reviewer 3**
>
> Response to con 1 and 2:
> Thanks for this comment! Due to space limit, choices of $h$ (model/classifier) are only included in our supplementary materials (Section C.6): for MNIST and Fashion MNIST, we use the convolutional neural network used in DMI [1,2]. For CIFAR-10 and CIFAR-100, we use an 18-layer PreAct Resnet model [3]. For Clothing 1M dataset, we use the pre-trained ResNet 50 model [3]. In experiment, as mentioned in appendix (Section A.2), we fixed variation function $\tilde{g}^*$ beforehand using an approximate activation function for each $f$ [4]. In Table 5 (appendix, Section A.2), $g_f^{v}$ is the form of $\tilde{g}^*$ (in Algorithm 1), while $f^*(u)$ is the choice of $f^*$ (in Algorithm 1). We will provide readers with more experiment details in our appendix and clearly mention this in Section 5.
> ----------------------------------
> Response to con 3:
> We follow the estimation procedure as being used in the literature. Due to space limitations, we could not reproduce all details for baseline methods. We will add it to the Appendix. Loss Correction [5] (BLC and FLC)} estimates the noise transition matrix via the following steps (we modify their notations accordingly by referring to our notations):
>
> (1) Given the noisy training dataset $\tilde{D}:=(X, \tilde{Y})$, any loss function $\ell$. Train a network $h(x)$ on $\tilde{D}$ with loss $\ell$.
>
> (2) Obtain an unlabeled sample $X'$ (in practice, their official code [6] chooses $X'=X$).
>
> (3) Find a perfect sample for each class $i$ where $i\in [K]$: $x^i_{per}=argmax_{x\in X'}\quad \hat{p}(\tilde{y}=i|x)$. In practice, the probability is calculated via the trained model $h(x)$'s prediction [6].
>
> (4) The estimated transition matrix is obtained by $T_{ij}=\hat{p}(\tilde{y}=i|x^i_{per})$.
> In practice, $T_{ij}$ needs to be normalized, or with some additional transformation as mentioned in [6].
> ----------------------------------
> Response to the minor issue:
> Thanks for pointing out! This is a typo. And we have corrected the generator function $f$ of KL divergence function: $f(v)=v\log v$.
> ----------------------------------
> Reference:
> 1. Xu, Y., Cao, P., Kong, Y., & Wang, Y. (2019). $L_{DMI}$: A Novel Information-theoretic Loss Function for Training Deep Nets Robust to Label Noise. In Advances in Neural Information Processing Systems (pp. 6225-6236).
> 2. https://github.com/Newbeeer/L_DMI/blob/master/fashion/fashion.py
> 3. He, K., Zhang, X., Ren, S., & Sun, J. (2016, October). Identity mappings in deep residual networks. In European conference on computer vision (pp. 630-645). Springer, Cham.
> 4. Nowozin, S., Cseke, B., & Tomioka, R. (2016). f-gan: Training generative neural samplers using variational divergence minimization. In Advances in neural information processing systems (pp. 271-279).
> 5.Patrini, G., Rozza, A., Krishna Menon, A., Nock, R., & Qu, L. (2017). Making deep neural networks robust to label noise: A loss correction approach. In Proceedings of the IEEE Conference on Computer Vision and Pattern Recognition (pp. 1944-1952).
> 6. https://github.com/giorgiop/loss-correction

---

### Official Review · AnonReviewer4 · 2020-10-28
**Good paper. A solid formulation and analysis of learning with noise labels by optimizing f divergence between predictions and labels. Includes both theoretical contribution and comprehensive empirical results.**

**Rating:** 7
**Confidence:** 3

**Review:**

This studies when maximizing a proper f divergence measure between the predicted labels and the observed labels would make the model robust with label noise. The authors derive that the f divergence can be decoupled into variational differences defined on distributions without noise and a bias term introduced by the noise. With this measure, we can learn with noisy labels without knowing the noise rate. They also consider the situations when the proposed f divergence measure is not robust against noise in labels, they propose a correction to mitigate this issue. Comprehensive experiments are done across several datasets with three noise models. Results show the proposed methods outperform baselines in most cases.

If the authors could explain the assumptions on the label noise that the work is based on early in the introduction and Section 2, the presentation would be improved. Now I have to read until Section 1.1 and Section 2.3 to know noise labels are generated based on the transition matrix.

In Table 2, in the sparse and high noise situation, the PL model outperforms the proposed method consistently. It would be better if the authors can discuss the difference between PL and the proposed method and provide reasons about why PL is better. Currently, the discussion focuses on why the proposed method cannot do well. In addition, the current explanations are conjectures. I would suggest the authors run experiments to verify such conjectures. For example, analyze the input of the g function to verify the first conjecture.

In Table 2, there are no results on random noise for MNIST and fashion MNIST. In addition, it did not distinguish Sparse Low and High and Uniform Low and High for CIFAR 100, could the authors explain this?

In Table 3, the J-S model with sparse high noise produces a gap value -20.88, which is way larger in magnitude than others. Could the authors further explain this result? It also lacks discussion on the meaning of the gap value, I wonder why such a large negative value has a trivial impact on the performance of the model.

---

> ### Author Response · Authors · 2020-11-13
> **Response to Reviewer 4**
>
> 1. Presentation issue w.r.t. assumptions on the label noise:
> Thanks for this suggestion! We will explain how noise labels are generated in the introduction section in our revised version.
> --------------------------
> 2. Comparison with PL:
> In practice, we think PL [1] is a 'special' case of the variational form: if we fix $\tilde{g}^*(v)=\log(v)$, and choose $f^*(u)=u$, maximizing this variational form of $``f"-$divergence is of the same optimization task as implemented in PL. Thus, if we consider PL as a special case of the variational form of a special $f-$divergence, we can explain why PL does not fall too behind from $f-$divergence functions. As shown in Table 2, our proposed method outperforms other related works in almost all cases except for the sparse high noise setting. We will explain this below.
> --------------------------
> 3. Experiment design and explanation for missing:
> We did not run the random noise setting for MNIST and Fashion MNIST because we thought these two datasets are of smaller scale and are generally easy to learn even under the noisy label settings. Experimentally, we didn't observe too much of a difference under different noise models. What is more, we think the random noise setting is somehow similar to a mixture of uniform (with a larger weight) noise and sparse noise.  We are currently running experiments on random noise setting for MNIST and Fashion MNIST and will update the experiment results in our revised version!
> Since our focus is mainly on the CIFAR-10 dataset (the most widely used one) and a real-world noise dataset Clothing 1M (the most challenging one, also the noise setting is feature dependent), we did not exhaustively test for all settings with CIFAR-100 (also due to the extensive amount of computation power needed). Technically, there are 100 categories in CIFAR-100, for the uniform noise model, the off-diagonal element $T_{ij}$ ($i\neq j$) of noise transition matrix are pretty small for both high and low noise rate setting. For example, $T_{ij}=0.01$ ($i\neq j$) will result in randomly assigning label for each image. Thus, we did not distinguish the high/low settings for CIFAR-100.
> --------------------------
> 4. Experiment issue w.r.t. sparse high noise setting:
> We shall revise our conjectures on the sparse high results accordingly. Thanks to the reviewer's suggestion we traced the inputs of the $g$ functions and found that for most of the bad training outcomes, the $g$ functions are actually receiving stable inputs, therefore negating one of our conjecture.
> As shown in Table 2, the sparse high noise setting tends out to be a super challenging task for all methods. We now think this might due to the fact that the sparse high noise might create a highly imbalanced dataset, which we have mentioned at the beginning of Section B in the appendix. For example, suppose one class pair $(i,j)$ ($i, j\in [K=10]$) in CIFAR-10 sparse high noise setting follows the following noise transition matrix:
> \\begin{bmatrix}
>     0.4 & 0.6\\\\
>     0.2 & 0.8
>     \\end{bmatrix} Thus, before the transition, the ratio of images between $i, j$ is $1:1$, however, after flipping labels, the ratio of images between $i, j$ becomes $3:7$.  The number of images in class $j$ is 2 times larger than that of $i$. As a result, an imbalanced noisy dataset makes training procedure unstable and results in bad performance frequently.

---

> > ### Author Response · Authors · 2020-11-13
> > **Additional explanation on the large gap**
> >
> >
> > As for the large gap that appeared in Table 3 for Jenson-Shannon, we further compared the training curve between JS (w/o bias correction) and JS (with bias correction). Even though these two settings are of the same hyper-parameter setting, the latter converges to sub-optimum at the very beginning, while the former one converges to a 'better' sub-optimum. Thus, we think the reason for this large gap attributes to a highly imbalanced training set as well as an unsuitable hyper-parameter setting (and the high variance and imbalance certainly have made the search for the best hyper-parameter harder).
> >
> > Even though we didn't exhaustively look for perfect hyper-parameter settings w.r.t each divergence in the sparse high noise setting, we happen to observe that if we apply an increasing weight to the conjugate term, maximizing weighted $f-$mutual information helps us avoid converging to a 'low-quality' sub-optimal hypothesis $h$ easily and returns a significantly better $h$. However, considering our theoretical conclusions and negligible effects of Bias term in other noise settings, we choose not to use the better result learned via weighted $f-$mutual information.
> > Thus, we think the reason for this large gap attributes to a highly imbalanced training set as well as an unsuitable hyper-parameter setting. Even though we didn't find out perfect hyper-parameter settings w.r.t each divergence in the sparse high noise setting, we happen to observe that if we apply an increasing weight to the conjugate term, maximizing weighted $f-$mutual information won't converge to 'low-quality' sub-optimum hypothesis $h$ easily and returns a significantly better $h$. However, considering our theoretical conclusions and negligible effects of Bias term in other noise settings, we choose not to use the better result learned via weighted $f-$mutual information.
> > -----------------------------------------------
> > Reference
> > 1. Liu, Y., & Guo, H. (2019). Peer Loss Functions: Learning from Noisy Labels without Knowing Noise Rates. arXiv preprint arXiv:1910.03231.

---

### Official Review · AnonReviewer1 · 2020-10-29
**Interesting work with solid experiments but some theoretical questions**

**Rating:** 6
**Confidence:** 3

**Review:**

The authors consider the problem of learning a classifier when the labels are noisy. They consider an approach where they optimize the mutual information between the predicted label and the noisy label (or more generally, a variant of mutual information defined by an f-divergence). To optimize over mutual information, they use the variational form of f-divergence. They show that their loss function is a biased version of the mutual information between the predictions and the clean labels. They argue that the bias term is small and that, for many choices of f-divergence, that the true optimal solution is optimal even in the noisy case over a specific subset. They then experimentally demonstrate that their approach performs better than the baselines for a large number of datasets and noise settings.

I found the paper interesting, however the theoretical justification is not fully clear to me. The authors should clarify the following two questions.

(1) Is f-mutual information a good loss function on clean data, for any choices of f-divergence? I see in Section 2.2 special cases for when this is true, but can f-mutual information return a very bad solution when these conditions are not true?

(2) In Theorem 8, why is $\mathcal{H}^*$ the only set that we need to worry about?

The authors use the variational form of f-divergence to optimize their loss function. Algorithmically, how is the term $g^*$ calculated and updated? Since $P$ and $Q$ are changing as the model is optimized, does $g^*$ need to be recalculated over time?

The experimental section shows compelling results compared to state-of-the-art approaches. I think including no noise as an additional comparison can improve the understanding of the theoretical questions from above.

Overall, I found the paper interesting with solid experimental results, though I still have some questions of the theoretical principles of the work.

---

> ### Author Response · Authors · 2020-11-13
> **Response to Reviewer 1**
>
> 1. How good is $h_f^*$ on clean data and additional experiments:
>     In Section 2.2, we give special cases when maximizing $f-$mutual information returns Bayes Optimal classifier. In general, it is not straightforward to quantify how bad the optimal classifier can be when these conditions are not satisfied. We are running experiments to test how good is $h_f^*$ when trained on clean data. We will update our experiment results in the revised paper once these experiments finished!
> ---------------------------------------------
> 2. Why is $\mathcal{H}^*$ the only set that we need to worry about:
> Our main purpose is to explore the robustness conditions of $f-$mutual information. One solution is to constrain the hypothesis space $\mathcal{H}$, as mentioned in Theorem 7 and 8, Total-Variation is $\mathcal{H}$-robustness, while the robustness hypothesis space for other divergences is constrained to be $\mathcal{H}^*$, where $\mathcal{H}^*$ contains the set of unfavorable classifiers that often are returned when the training overfits to the noisy label distribution. This is because the Bias term avoids $h_f$ from converging to $h_f^*$} unless $h$ is in the set we need to worry about. However, in Section 4.1, we theoretically validated the negligible effect of the Bias term. And our experiment results (Table 3) also demonstrate that in most noise settings, the impact of bias term can be ignored.
> ---------------------------------------------
> 3. Calculation of $g^*$:
> In experiment, we fixed variation function $\tilde{g}^*$ beforehand using an approximate activation function for each $f$ [1]. In Table 5 (appendix, Section A.2), $g_f^{v}$ is the form/choice of $\tilde{g}^*$ (in Algorithm 1). For example, if we choose Total-Variation, we can just fix $\tilde{g}^*(v)=\dfrac{1}{2}\tanh(v)$ (in Algorithm 1, step 4). And we do not need to update or recalculate the form of $\tilde{g}^*$. We will clarify this in Section 5.
> The reviewer is absolutely correct that the inputs to the $\tilde{g}$ function will change, because $P$ and $Q$ are defined w.r.t. the model's prediction and the noisy label distribution. At each epoch of our training, we will need to update $P$ and $Q$ using a sub-batch of the training data. In experiments, due to limitations of computing resources, we choose the batch size to be 16 for Clothing 1M. For other datasets, the batch size is 128.
> ---------------------------------------------
> Reference:
> 1. Nowozin, S., Cseke, B., & Tomioka, R. (2016). f-gan: Training generative neural samplers using variational divergence minimization. In Advances in neural information processing systems (pp. 271-279).

---

### Official Review · AnonReviewer2 · 2020-10-31
**good paper. I have just few minor comments.**

**Rating:** 7
**Confidence:** 2

**Review:**

##########################################################################
Summary:

The manuscript identify robustness properties of the f-divergence optimization when noisy labels presents. The f-divergence measure is defined between the joint distribution and the marginal distributions of classifier’s predictions and labels. In this case, the f-divergence measure can be interpreted as the f-mutual information and thus the optimal classifier can be estimated by maximizing the f-divergence measure. The authors examine a family of f-divergence functions (generators in f-divergence measures) and derive several interesting robustness properties. For example, total variation is robust for the label noise problem (the label noise does not affect the optimality of the maximizer; it does not require the knowledge of noise rate), and when the f-divergence functions are not robust, we can still correct the solution to be the optimal by calculating the bias term introduced by the noisy labels.

##########################################################################
Reasons for score:

I vote for accepting. The manuscript provides theoretical analyses of the f-divergence optimization with noisy labels. Although there are several research works on training with f-divergence measures recently, I can see the novelty of this work regarding the robustness of the f-divergence optimization when noisy labels presents. The effectiveness of optimizing f-divergence with label noise is also verified by a set of numerical experiments. I think that this work can be a good reference about robust training approaches dealing with label noise.

##########################################################################
Suggestions
I have just few minor suggestions.
1)	At first glance, maximization of f-divergence measures sounds confusing in the abstract and introduction as we generally minimize any divergence measures to train a model to fit the training data. I think that it would be better if the authors include additional explanations (in the abstract and introduction) that the f-divergence measure can be understood as the f-mutual information in this problem formulation.
2)	Below equation 2), f(v)=log v should be f(v)=v log v

---

> ### Author Response · Authors · 2020-11-13
> **Response to Reviewer 2**
>
> Response to minor suggestion 1:
> Thanks for this suggestion! Indeed to not confuse our readers, we use "optimizing" rather than "maximizing" in our title. We will add some additional explanations of $f-$mutual information in the abstract and introduction in our revised version (will be uploaded once additional experiments finished).
> ---------------------------------------------
> Response to minor suggestion 2:
> Thanks for pointing out! This is a typo. And we have corrected the generator function $f$ of the KL divergence function. The correct form should be: $f(v)=v\log v$.

---

### Author Response · Authors · 2020-11-19
**Revision of our paper and appendix are uploaded!**

Dear Reviewers,

We thank all reviewers for their thoughtful and helpful comments! We have uploaded our revised main paper and appendix. Changes with respect to the previous version are highlighted in light-blue. To summarize, we revise the following in our paper:

1.  In our abstract, we mention that the f-divergence measure can be understood as $f-$mutual information (Suggested by Reviewer 2).
2. We corrected the generator function of the KL divergence function (Mentioned by Reviewer 2 and Reviewer 3).
2.  We moved the explanations and assumptions on the label noise to the beginning of Section 2 (Suggested by Reviewer 4).
3. We added experiment details of $\tilde{g}^*$ in Algorithm 1 (Mentioned by Reviewer 1 and Reviewer 4).
4. We added Section 5.1, to illustrate how good is $h_f^*$ on the clean datasets of  MNIST, Fashion MNIST, CIFAR-10, CIFAR-100. Experiment results show that maximizing $f-$ mutual information on the clean dataset returns a promising classifier by referring to the Cross-Entropy loss. Please refer to the newly added Table 2 for details (Suggested by Reviewer 1).
5. We added experiment results on Random noise in MNIST and Fashion MNIST where $D_f$ measures outperform all baseline methods (Mentioned by Reviewer 4).
6. We revised our conjectures on the performance of the Sparse High noise setting (Mentioned by Reviewer 4).
7. We added experiment details of $D_f$ measures on the clean dataset in the appendix, section C.7.

Best

Authors

---

### Decision · Program_Chairs · 2021-01-07
**Final Decision**

**Decision:**

Accept (Poster)

**Comment:**

The paper tackles a very important problem. The formulation of the paper is sound as under lightweight assumptions, the supervised loss follows an f-divergence formulation (see "Information, Divergence and Risk for Binary Experiments" by Reid and Williamson (JMLR 2011), in particular Section 4.7). It would make sense to dig in the loss in the context of label noise; the variational formulation provides an interesting direction along those lines. The rebuttal on the experimental concerns of reviewers is appreciated (Cf authors’ rebuttal summary).